# Neurocoder: Learning General-Purpose Computation Using Stored Neural Programs

## Abstract

Artificial Neural Networks are functionally equivalent to special-purpose comput-
ers. Their inter-neuronal connection weights represent the learnt Neural Program
that instructs the networks on how to compute the data. However, without stor-
ing Neural Programs, they are restricted to only one, overwriting learnt programs
when trained on new data. Here we design Neurocoder, a new class of general-
purpose neural networks in which the neural network "codes" itself in a data-
responsive way by composing relevant programs from a set of shareable, modular
programs stored in external memory. For the first time, a Neural Program is ef-
ficiently treated as a datum in memory. Integrating Neurocoder into current neu-
ral architectures, we demonstrate new capacity to learn modular programs, reuse
simple programs to build complex ones, handle pattern shifts and remember old
programs as new ones are learnt, and show substantial performance improvement
in solving object recognition, playing video games and continual learning tasks.

## 1 Introduction

From its inception in 1943 until recently, the fundamental architectures of Artificial Neural Net-
works remained largely unchanged - a program is executed by passing data through a network of
artificial neurons whose inter-neuronal connection weights are learnt through training with data.
These inter-neuronal connection weights, or Neural Programs, correspond to a program in modern
computers [32]. Memory Augmented Neural Networks (MANN) are an innovative solution allow-
ing networks to access external memory for manipulating data [11, 12]. But they were still unable
to store Neural Programs in such external memory, and this severely limits machine learning. Stor-
ing inter-neuronal connection weights only in their network does not permit modular separation of
Neural programs and is analogous to a computer with one fixed program. Recent works introduce
*conditional computation* via adjusting or activating parts of a network in an input-dependent manner
[39, 33, 4, 13, 28], but networks remain monolithic. Current networks forget when retrained, old
inter-neuronal connection weights are merged with new ones or erased.

The brain is modular, not a monolithic system [8, 6]. Neuroscience research indicates that the brain is
divided into functional modules [19, 7, 9]. If the neural program for each module is kept in separate
networks, networks proliferate. Modular neural networks, another form of conditional computation,
combine the output of multiple expert networks, but as the experts grow, the networks grow drasti-
cally [20, 14, 35, 29]. This requires huge computational storage and introduces redundancy as these
experts do not share common basic programs.

*A pathway out of this bind is to keep such basic programs in memory and combine them as required.*
This brings neural networks towards modern general-purpose computers that use the stored-program
principle [37, 40] to efficiently access reusable programs in external memory. Here we show how
Neurocoder, a new neural framework, introduces a new class of general-purpose conditional compu-

Submitted to 35th Conference on Neural Information Processing Systems (NeurIPS 2021). Do not distribute.

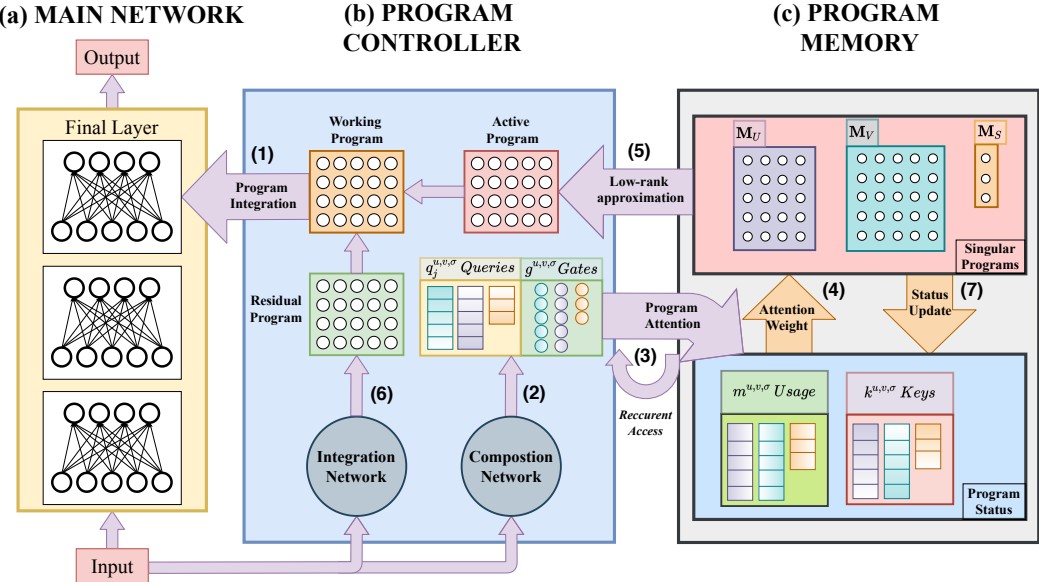

Figure 1: Neurocoder **(a)** The *Main Network* uses a *working program* to compute the output for the input. Here only the final layer of the Main Network is adaptively loaded with the working program **(1)**. Other layers use traditional Neural Programs as connection weights (fixed-after-training). **(b)** The Program Controller's *composition network* controls access to the Program Memory, emitting queries and interpolating gate control signals in response to the input **(2)**. It then performs recurrent multi-head program attention to the Program Status **(3)**, triggering attention weights to the Singular Programs **(4)**. The attended Singular Programs form an *active program* using low-rank approximation **(5)**. *Residual program* produced by the Program Controller's *integration network* **(6)** plus the active program derives the *working program*. **(c)** The Program Memory stores the representations (singular programs) required to reconstruct the active program to be used by the Program Controller. Access is controlled through the Program Status including keys ($k$), and slot usage ($m$) that are updated during the training and computation **(7)**.

tation machines in which a neural network can be "coded" in an input-dependent manner. Efficient decomposition of Neural Programs creates shareable modular components that can reconstruct the whole program space. These components change their "shapes" based on training and are stored in an external Program Memory. Then, in a data-responsive way, a Program Controller retrieves relevant components to build the Neural Program. This is analogous to shape-shifting Lego bricks that can be reused to build unlimited shapes and structures (See Appendix Fig. 4).

Using adaptive modular components vastly increases the learning capacity of the neural network by allowing re-utilisation of parameters, effectively curbing network growth as programs increase. More importantly, unlike pre-defined sub-networks or modules [20, 1] that combine at activation level, the construction of our modular components is dynamic and performed on the weight space. The Neural Program construction is learnt through training via traditional backpropagation [30] as the architecture is end-to-end differentiable.

## 2 Methods

### 2.1 System overview

A Neurocoder is a neural network (Main Network) coupled to an external Program Memory through a Program Controller. The *working program* of the Main Network processes the input data to produce the output. This working program is "coded" by the Program Controller by creating an input-dependent *active program* from the Program Memory (Fig. 1). The following gives a high-level description of the Neurocoder framework and then the details.

**Neurocoder stores Singular Value Decomposition of Neural Programs in Program Memory**

The Neural Program needs to be stored efficiently in Program Memory. This is challenging as there may be millions of inter-neuronal connection weights, thus storing them directly ([22]) is grossly inefficient. Instead, the Neurocoder forms the basis of a subspace spanned by Neural Programs and stores the singular values and vectors of this subspace in memory slots of the Program Memory (hereafter referred to as *singular program*s). Based on the input, relevant singular programs are retrieved, a new program is reconstructed and then loaded in the Main Network to process the input. This representational choice significantly reduces the number of stored elements and allows each singular program to effectively represent a unitary function of the active program.

The *active program* matrix $\mathbf{P}$ can be composed by standard low-rank approximation as

$$\mathbf{P} = \mathbf{USV^T} = \sum_n^{r_m} \sigma_n u_n v_n^\top \tag{1}$$

where $\mathbf{U}$ and $\mathbf{V}$ are matrices of the left and right singular vectors, and $\mathbf{S}$ the matrix of singular values. $r_m$ is the total number of components we want to retrieve. $\{\sigma_n\}_{n=1}^{r_m}$ is the attended singular values, $\{u_n\}_{n=1}^{r_m}$ and $\{v_n\}_{n=1}^{r_m}$ the attended singular vectors of $\mathbf{S}$, $\mathbf{U}$, and $\mathbf{V}$, respectively. The Program Memory is crafted as three *singular program memories* $\{\mathbf{M}_U, \mathbf{M}_V, \mathbf{M}_S\}$–each of their memory slot stores a singular component or singular program. *The process "codes" the active program using singular programs from* the program memories. The coding is conditioned on input $x_t$, yet we drop index $t$ for notation simplification and leave the details on the computation of $\sigma_n, u_n, v_n$ in Sec. 2.2.

The Program Memory also maintains the status for each singular program in terms of access and usage. To access a singular program, *program keys* ($k$) are used. These keys are low-dimensional vectors that represent the singular program function and computed by a neural network that effectively compresses the singular program. The *program usage* ($m$) measures memory utilisation, recording how much a memory slot is used in constructing a program. The components of the Program Memory are summarised in Fig. 1 (c).

**Recurrent multi-head program attention mechanisms for program storage and retrieval**
Neural networks use the concept of *differentiable attention* to access memory [11, 2]. This defines a weighting distribution over the memory slots essentially weighting the degree to which each memory slot participates in a read or write operation. This is unlike conventional computers that use a unique address to access a single memory slot.

Here we use two kinds of attention. First is *content-based attention* [11, 12] to ensure that the singular program is selected based on its functionality and the data input. This is achieved by producing a query vector based on the input and comparing it to the program keys ($k$) using cosine similarity. Higher cosine similarity scores indicate higher attention weights to the singular programs associated with those program keys. Second, to encourage better memory utilisation, higher attention weights are assigned to slots with lower program usage ($m$) through *usage-based attention* [12, 31]. The attention weights from the two schemas are then combined using interpolating gates to compose the final attention weights to the Program Memory.

We adapt multi-head attention [11, 38] that applies multiple attentions in parallel to retrieve $H$ singular components. Besides, we introduce a recurrent attention mechanism, in which multi-head access is performed recurrently in $J$ steps. The $j$-th set of $H$ retrieved components is conditioned on the previous ones. This recurrent, multi-head attention allows the composition network to incrementally search for optimal components for building relevant active programs.

**Neurocoder learns to "code" a relevant working program via training**

The structure of the Program Memory and the role of the Program Controller facilitates the automatic construction of working programs via training. The Program Controller controls memory access through its *composition network* that creates the *attention weight* defining how to weight the singular programs in the memories. A weighted summation of the singular programs results in the attended singular program. Applying the recurrent multi-head attention described earlier, multiple attended singular programs are retrieved to construct an active program (Eq. 1). Then the Program Controller generates a *residual program* using its *integration network*, adding to the active program

to produce the working program of the Main Network. This addition enables creation of flexible higher-rank working programs, which compensates for the low-rank coding process. The structure of the Program Controller is illustrated in Fig. 1 (b).

The singular programs are trained to represent unitary functions necessary for any computation whilst the composition and integration networks are trained to compose the relevant programs for the considering task. As such, beside minimising the task loss, we enforce orthogonality of stored singular vectors by minimising $\mathcal{L}_o = \mathbf{M}_U \mathbf{M}_U^\top - \mathbf{I} + \mathbf{M}_V \mathbf{M}_V^\top - \mathbf{I}$. The parameters of the networks, and the stored singular programs are adjusted using gradient training via minimising the total loss

$$\mathcal{L} = \mathcal{L}_{task} + a\mathcal{L}_o \tag{2}$$

where $\mathcal{L}_{task}$ represents the supervised task loss and $\mathcal{L}_o$ represents the orthogonal loss weighted by a hyper-parameter $a$ to enforce orthogonality of the singular vectors.

## 2.2 Attention mechanisms for Program Memory

Here we describe program attention mechanisms used in this paper. Given $w_{in}^u$, $w_{in}^v$, $w_{in}^\sigma$ (jointly denoted as $w_{in}^{u,v,\sigma}$)–the attention weight to the $i$-th slot of the singular program memories $\mathbf{M}_U$, $\mathbf{M}_V$ and $\mathbf{M}_S$, we retrieve the $n$-th singular vector as follows,

$$u_n = \sum_{i=1}^{P_u} w_{in}^u \mathbf{M}_U(i) \tag{3}$$

$$v_n = \sum_{i=1}^{P_v} w_{in}^v \mathbf{M}_V(i) \tag{4}$$

For the singular values, we need to enforce $\sigma_1 > \sigma_2 > ... > \sigma_{r_m} > 0$, thus we retrieve using

$$\sigma_n = \begin{cases} \text{softplus}\left(\sum_{i=1}^{P_s} w_{in}^\sigma \mathbf{M}_S(i)\right) & n = r_m \\ \sigma_{n+1} + \text{softplus}\left(\sum_{i=1}^{P_s} w_{in}^\sigma \mathbf{M}_S(i)\right) & n < r_m \end{cases} \tag{5}$$

Here, $P_u$, $P_v$ and $P_s$ are the number of memory slots of $\mathbf{M}_U$, $\mathbf{M}_V$ and $\mathbf{M}_S$, respectively. In this paper, we set $P = P_u = P_v = P_s$ as the number of memory slots of the Program Memory. We note that these notations are specified for some data input $x_t$ and the index $n$ later maps to an attention head $h$, and an attention step $j$, hence the full notation should be $w_{tijh}^{u,v,\sigma}$. To simplify notations, we will drop $u, v, \sigma$ from now and describe the computation of a representative $w_{tijh}$ for any of the three program memories in the following parts.

### Recurrent Access to the Program Memory via the composition network

To perform program attention, the Program Controller employs a composition network (denoted as $f_\theta$), which takes the current input $x_t$ and produce *program composition control signals* ($\boldsymbol{\xi}_t^p$). If $f_\theta$ performs all attentions concurrently via multi-head attention as in [11, 38], it may lead to program collapse [22]. To have a better control of the component formation and alleviate program collapse, we propose to recurrently attend to the program memory. To this end, we implement $f_\theta$ as a recurrent neural network (LSTM [16]) and let it access the program memory $J$ times, resulting in $\boldsymbol{\xi}_t^p = \left\{\boldsymbol{\xi}_{tj}^p\right\}_{j=1}^J$. At access step $j$, the recurrent network updates its hidden states and generates $\boldsymbol{\xi}_{tj}^p$ using recurrent dynamics as

$$\boldsymbol{\xi}_{tj}^p, h_j = f_\theta(x_t, h_{j-1}) \tag{6}$$

where $h_0$ is initialized as zeros and $\boldsymbol{\xi}_{tj}^p$ is the program composition control signal at step $j$ that depends on both on the input data $x_t$ and the the previous state $h_{j-1}$. Particularly, the control signal contains the queries and the interpolation gates for each head to compute the program attention weight: $\boldsymbol{\xi}_{tj}^p = \{q_{tjh}, g_{tijh}\}_{h=1}^H$. Here, at each attention step, we perform multi-head attention with $H$ as the number of attention heads and thus, each $\boldsymbol{\xi}_{tj}^p$ consists of $H$ pairs of queries and gates. Hence, the total number of retrieved components $r_m = J \times H$ and the index $n = j \times H + h$.

**Attending to Programs by "Name"**

Inspired by the content-based attention mechanism for data memory [11], we use the query to look for the singular programs. In computer programming, to find the appropriate program for some computation, we often refer to the program description or at least the name of the program. Here, we create the "name" for our neural programs by compressing the program content to a low-dimensional key vector. As such, we employ a neural network ($f_\varphi$) to compute the program memory keys as

$$k_i = f_\varphi \left( \mathbf{M} \left( i \right) \right) \tag{7}$$

where $k_i \in \mathbb{R}^K$ and $i$ is the row index of the program memory. Here, $f_\varphi$ learns to compress each memory slot into a $K$-dimensional vector. As the singular programs evolve, their keys get updated. In this paper, we update the program keys after each learning iteration during training.

Finally the content-based program memory attention $c_{tijh}$ is computed using cosine distance between the program keys $k_i$ and the queries $q_{tjh}$ as

$$c_{tijh} = \text{softmax}^{(i)} \left( \frac{q_{tjh} \cdot k_i}{||q_{tjh}|| \cdot ||k_i||} \right) \tag{8}$$

**Making Every Program Count**

Similarly to [12, 31], in addition to the content-based attention, we employ a least-used reading strategy to encourage the Program Controller to assign different singular programs to different components. In particular, we calculate the memory usage for each program slot across attentions as

$$m_{tijh} = \max_{\tilde{j} \leq j} \left( w_{ti\tilde{j}h} \right) \tag{9}$$

Since we want to consider only $l_I$ amongst $P$ memory slots that have smallest usages, let $\hat{m}_{tjh}^{l_I}$ denote the value of the $l_I$-th smallest usage, then the least-used attention is computed as

$$l_{tijh} = \begin{cases} \max\limits_{i} \left( m_{tijh} \right) - m_{tijh} & ; m_{tijh} \leq \hat{m}_{tjh}^{l_I} \\ 0 & ; m_{tijh} > \hat{m}_{tjh}^{l_I} \end{cases} \tag{10}$$

The final program memory attention is computed as

$$w_{tijh} = \text{sigmoid} \left( g_{tijh} \right) c_{tijh} + \left( 1 - \text{sigmoid} \left( g_{tijh} \right) \right) l_{tijh} \tag{11}$$

Since the usage record are computed along the memory accesses, the multi-step Neurocoder utilises this attention mechanism better than the single-step Neurocoder, creating different attention styles (see Sec. 3.2). The composition the active program $\mathbf{P}_t$ is illustrated in Appendix's Fig. 5.

### 2.3  Program Integration via the integration network

Since the working program $\mathbf{P}_t$ only contains top $r_m$ principal components, it is low-rank and may be not flexible enough for sophisticated computation. We propose to enhance $\mathbf{P}_t$ with a residual program $\mathbf{R}$– a traditional connection weight trained as the integration network's parameters, which is constant after training w.r.t $t$. The residual program represents the sum of the remaining less important components. To this end, we suppress $\mathbf{R}$ with a multiplier that is smaller than $\sigma_{tr_m}$– the smallest singular value of the main components - resulting in the integration formula

$$W_t = \mathbf{P}_t + w_t^r \sigma_{tr_m} \mathbf{R} \tag{12}$$

where $w_t^r = \text{sigmoid} \left( f_\phi \left( x_t \right) \right)$ is an adaptive gating value that controls the contribution of the residual program. $f_\phi$ is the integration network in the Program Controller and hence, in our implementation, the integration control signal sent by the Program Controller is $\boldsymbol{\lambda}_t^p = \{w_t^r, \sigma_{tr_m}\}$. We note that in our experiments, the program integration can be disabled ($W_t$ is directly set to $\mathbf{P}_t$) to prove the contribution of $\mathbf{P}_t$ or reduce the number of parameters. The working program $W_t$ is then used by the Main Network to execute the input data $x_t$ (see (Fig. 1 (a))). For example, with linear

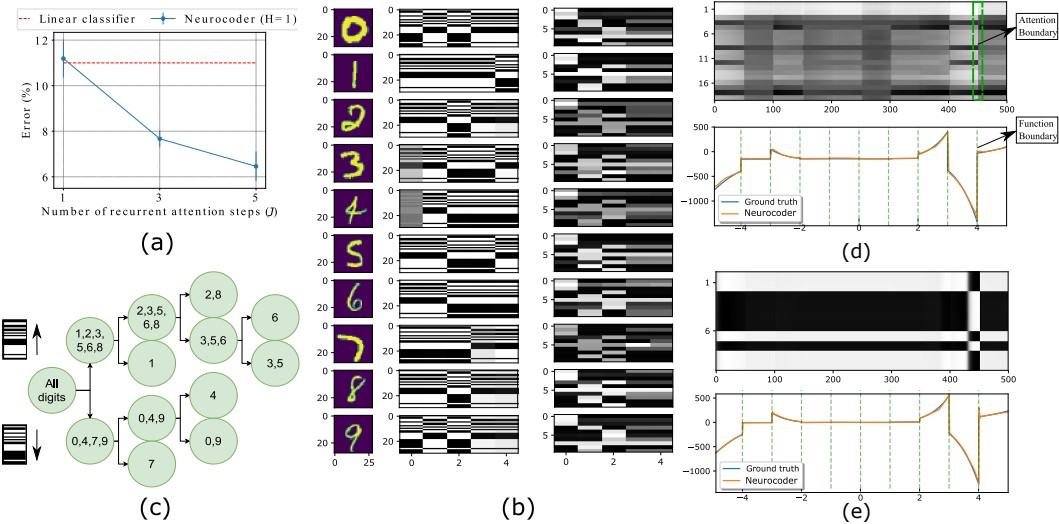

Figure 2: **(a)** MNIST test set classification error *vs* the number of steps ($J$) in Neurocoder (blue), compared with a linear classifier (red). **(b)** *1st column*: Digit images; *Middle column*: Single-step attention weights for 30 slots in $\mathbf{M}_U$ (vertical axis) for first 3 singular vectors (horizontal axis) for each digit; *Last column*: Multi-step attention weights for 10 slots in $\mathbf{M}_U$ (vertical axis) for first 3 singular vectors (horizontal axis). Multi-step attention is able to produce far more diverse patterns with fewer slots - 10 slots compared to single-step 30 slots. **(c)** Two attention patterns of single-step Neurocoder. The binary decision tree derived from single-step Neurocoder's attention patterns. The two patterns across components represent the decisions going up and down across the binary tree. Visualisation for **(d)** multi-step ($J = 5$, 20 memory slots) and **(e)** single-step ($J = 1$, 10 memory slots) cases showing while processing a sequence of the polynomial auto-regression task. The Neurocoder's attentions to $\mathbf{M}_U$ that form the first component of the active program are shown over sequence timesteps (*upper*) with Neurocoder's $y_t$ prediction (orange) and ground truth (blue) (*lower*). The vertical dash green lines separate polynomial chunks. Each chuck represents a local pattern, and thus ideally requires a specific active program to compute the input $x_t$. Although both predict well, only the multi-step Neurocoder discovers the chunk boundaries, assigning program attention to the first component in accordance with sequence changes.

classifier Main Network, the execution is $y_t = x_t W_t$. Appendix's Table 2 summarises the notations used for important parameters of Neurocoder.

# 3 Results

To demonstrate the flexibility of Neurocoder framework, we consider different learning paradigms: instance-based, sequential, multi-task and continual learning. We do not focus on breaking performance records by augmenting state-of-the-art models with Neurocoder. Rather our inquiry is on re-coding feed-forward layers with the Neurocoder's programs and testing on varied data types to demonstrate its intrinsic properties. For some experiments, we include ablation studies.

We compare the performance of diverse Main Networks (MN) with and without Neurocoder. We also augment the Main Networks with other recent conditional computing methods, either modular (sparse Mixture of Experts, Neural Stored-program Memory) or monolithic (HyperNets, FiLM) to form stronger baselines across our experiments. In our experiments, we always apply Neurocoder to all layers of multi-layer perceptrons (MLP) or just the final feed-forward layer of deep CNN networks (LeNet, DenseNet, ResNet), RNNs (GRU, LSTM), MANN (NTM). Other competitors such as MOE, NSM, HyperNet and FiLM are applied to the Main Networks in the same manner.

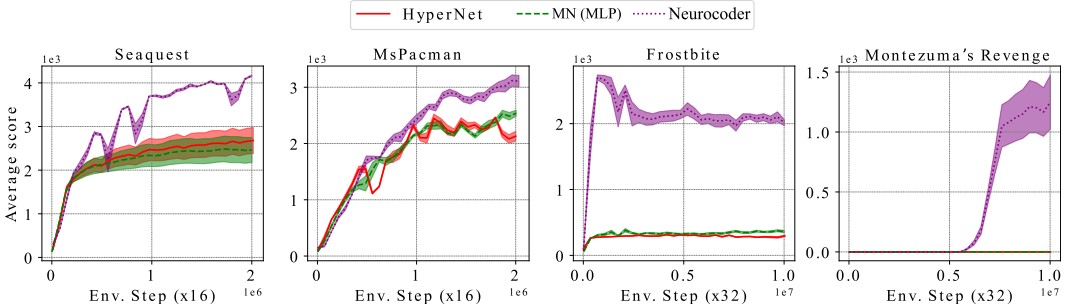

Figure 3: Learning curves (mean and std. over 5 runs) on representative Atari 2600 games. All baselines are applied to the actor/critic networks in the A3C agent.

## 3.1 Instance-based learning - Object Recognition

We tested Neurocoder on instance-based learning through classical image classification tasks using MNIST [24] and CIFAR [21] datasets. The first experiment interpreted Neurocoder's behaviour in classifying digits into 10 classes ($0 - 9$) using linear classifier Main Network. With equivalent model size, Neurocoder using the novel recurrent attention surpasses the performance of the linear classifier [24] by up to $5\%$ (Fig. 2 (a)).

To differentiate the input, Neurocoder attends to different components of the active program to guide the decision-making process. Fig. 2 (b) shows single-step and multi-step attention to the first 3 singular vectors for each digit across memory slots. Multi-step attention produces richer patterns compared to single-step Neurocoder that manages only 2 attention weight patterns.

Fig. 2 (c) illustrates how Neurocoder performs modular learning by showing the attention assignment for top 3 singular vectors as a binary decision tree. Digits under the same parental node share similar attention paths, and thereby similar active programs. Some digits look unique (e.g. 7) resulting in active programs composed of unique attention paths, discriminating themselves early in the decision tree. Some digits (e.g. 0 and 9) share the same attention pattern for the first 3 components and are thus unclassifiable. They can only be distinguished by considering more singular vectors.

We integrated Neurocoder with deep networks - 5-*layer LeNet and* $100$-*layer DenseNet* - and tested on CIFAR datasets. Neurocoder significantly outperformed the original Main Networks with performance gain $1-5\%$. Compared with recent conditional computing models such as sparse Mixture of Experts (MOE [35]) and Neural Stored-program Memory (NSM [22]), Neurocoder required a tenth of the number of parameters and performed better by up to $8-10\%$ (see Appendix's Table 3).

## 3.2 Sequential learning - Adaption to sequence changes and game playing using reinforcement learning

Recurrent neural networks (RNN) can learn from sequential data by updating the hidden states of the networks. However, this does not suffice when local patterns shift, as is often the case. We now demonstrate that Neurocoder helps RNNs overcome this limitation by composing diverse programs to handle sequence changes.

**Synthetic polynomial auto-regression** We created a simple auto-regression task in which data points are sampled from polynomial function chunks that change over time. The Main Network is a strong *RNN–Gated Recurrent Unit* (*GRU* [5]). We found that GRU integrated with a single-step or multi-step Neurocoder converged much faster than all other baselines. The other conditional computing counterparts (HyperNet [13], FiLM [28]) adapt by re-scaling weights or activation of the GRU, which were shown inferior to our modular approach (Appendix's Fig. 6).

Visualising the first singular vector attention weights in $\mathbf{M}_U$, we find that the multi-step attention Neurocoder changes its attention following polynomial changes - it attends to the same singular program when processing data from the same polynomial and alters attention for data from a different polynomial (Fig. 2(d)). In contrast, the single-step Neurocoder only changes its attention when there is a remarkable change in $y$-coordinate values (Fig. 2(e)). Although single-step Neurocoder

| Method | MN (MLP [17]) | MN (MLP ours) | NSM | Neurocoder |
|--------|---------------|---------------|-----|------------|
| Adam | 55.16±1.38 | 53.55±1.27 | 54.85±2.81 | **58.46±0.46** |
| Adagrad | 58.08±1.06 | 57.83±2.74 | 58.42±1.87 | **62.28±4.03** |
| L2 | 66.00±3.73 | 64.37±2.40 | 62.83±7.21 | **69.89±1.72** |
| SI | 64.76±3.09 | 64.41±3.36 | 64.36±2.99 | **67.96±3.22** |
| EWC | 58.85±2.59 | 58.41±2.37 | 58.12±3.24 | **65.66±1.25** |
| O-EWC | 57.33±1.44 | 57.78±1.84 | 58.55±3.40 | **73.97±1.50** |

Table 1: Incremental domain continual learning with Split MNIST. Final test accuracy (mean and std.) over 10 runs.

converges well, it did not discover the underlying structure of the data, and thus underperformed the multi-step Neurocoder. We hypothesise that when recurrence is employed, usage-based attention takes effect, stipulating better memory utilisation and diverse attentions over timesteps. We ran multi-step Neurocoder without usage-based attention. The results were worse than the full multi-step Neurocoder, which confirms our hypothesis (Appendix's Fig. 6).

**Atari game reinforcement learning**    We used reinforcement learning as a further testbed to show the ability to adapt to environmental changes. We performed experiments on several Atari 2600 games [3] wherein the agent was implemented as the *Asynchronous Advantage Actor-Critic (A3C* [26]). In the Atari platform, agents are allowed to observe the screen snapshot of the games and act to earn the highest score. We augmented the A3C by employing Neurocoder's working programs for feed-forward layers of the actor and critic networks, aiming to decompose the policy and value function into singular programs that were selected depending on the game state.

*Frostbite and Montezuma's Revenge.* These games are known to be challenging for A3C and other algorithms [26]. We trained A3C and HyperNet-based A3C for over 300 million steps, yet these models did not show any sign of learning, performing equivalently to random agents. For such complicated environments with sparse rewards, both the monolithic neural networks and the HyperNet's unstored fast-weights fail to learn (almost zero scores). In contrast, Neurocoder enabled A3C to achieve from $1,500$ to $3,000$ scores on these environments (Fig. 3), confirming the importance of decomposing a complex solution to smaller, simple stored programs.

### 3.3   Multi-task learning - Solving mutliple algorithms simultenously

Here we explore the modular learning capability of Neurocoder in multi-task setting. Inspired by algorithmic sequencing tasks [22], we created a challenging sequential multi-task benchmark wherein the input sequence is a series of sub-sequences from 4 algorithms: Copy, Repeat Copy, Associative Recall and Priority Sort [11]. Each sub-sequence, following a task identification vector, represents the input for each task. In each input sequence, $n$ tasks were sampled from the set of 4 algorithms randomly with replacement and the output sequences were created correspondingly.

We trained a *MANN–Neural Turing Machine (NTM* [11]*)* Main Network with FiLM, HyperNet and our Neurocoder augmentation on sequences of $n = 4$ tasks, and tested with sequences of $n = 4$ and $n = 8$ tasks. Appendix's Fig. 7 demonstrates that Neurocoder was performant in both test settings, not only achieving lowest error on $n = 4$, but also being the only one generalised well to $n = 8$ scenario, which was unseen during training.

### 3.4   Continual learning - Learning tasks sequentially without catastrophic forgetting

In continual learning, standard neural networks often suffer from "catastrophic forgetting" in which they cannot retain knowledge acquired from old tasks upon learning new ones [10]. Our Neurocoder offers natural mitigation of such catastrophic forgetting in neural networks by attending to different singular programs whilst learning different tasks.

In this case, in addition to the Main Network, we examine several continual learning algorithms with and without Neurocoder. These algorithms, including Elastic Weight Consolidation (EWC [41]) and Synaptic Intelligence (SI [41]), work by regularising the loss function and thus can be easily combined with Neurocoder by modifying the loss $\mathcal{L}_{task}$. We demonstrate that Neurocoder

can improve these continual learning algorithms without requiring additional assumptions as in other approaches [25, 36, 34] that either utilise task embedding or replay memory.

**Split MNIST** We first considered the split MNIST dataset–a standard continual learning benchmark wherein the original MNIST was split into a 5 2-way classification tasks, consecutively presented to a *Multi-layer Perceptron* Main Network (MLP). We followed the benchmarking as in [17] in which various optimisers and state-of-the-art continual learning methods were examined under incremental task and domain scenarios. We measured the performance of the MLP versus Neurocoder and NSM under each continual learning method. In both scenarios, Neurocoder was compatible with all continual leaning methods, demonstrating superior performance over MLP and NSM with performance gain between 1 to 16% (see Appendix's Table 5 and 1).

**Split CIFAR** We verified the scalability of Neurocoder to more challenging datasets. We split CIFAR datasets as in the split MNIST, resulting in 5-task 2-way split CIFAR10 and a 20-task 5-way split CIFAR100. We used Main Network *ResNet* [15]–a very deep CNN architecture.

When we stressed the orthogonal loss ($a = 10$) and used bigger program memory (100 slots), Neurocoder improved ResNet classification by 15% and 10% on CIFAR10 and CIFAR100, respectively. When we integrated Neurocoder with Synaptic Intelligence (SI [41]), the performance was further improved, maintaining a stable performance above 80% accuracy for CIFAR10 and outperforming using SI alone by 10% for CIFAR100 (see Appendix's Fig. 8).

## 4 Discussion

Our experiments demonstrate that Neurocoder is capable of re-coding Neural Programs in distinctive neural networks, amplifying their capabilities in diverse learning scenarios: instance-based, sequential, multi-task and continual learning. This consistently results in significant performance increase, and further creates novel robustness to pattern shift and catastrophic forgetting. This ability for each architecture to re-code itself is made possible without changing the way it is trained, or majorly increasing the number of parameters it needs to learn (see Appendix Table 7).

The MNIST problem illustrates the reasoning process of Neurocoder when classifying digit images wherein its singular program assignment resembles a binary tree decision-making process - it shows how some singular programs are shared, others are not. The polynomial auto-regression problem highlights the importance of efficient memory utilisation in re-constructing the working program enabling discovery of hidden structures in sequential data. Training our framework with reinforcement learning, we enable neural agents to solve complex games wherein traditional methods fail or learn slowly. Neurocoder also works well with multi-task setting, as shown in the challenging multi-algorithm benchmark. Finally, continual learning problems show that Neurocoder mitigates catastrophic forgetting efficiently under different learning settings/algorithms.

Our solution offers a single framework that is scalable and adaptable to various problems and learning paradigms. Unlike previous attempts to employ a bank of separate big programs [20, 35, 22], Neurocoder maintains only shareable, smaller components that can reconstruct the whole program space, thereby heavily utilising the parameters and preventing the model from proliferating. We note that Neurocoder is orthogonal to approaches employing tensor decomposition to reduce the number of parameters or hasten the computation [27, 23]. Neurocoder composes rather than decompose the neural weights. Our aim is not only to enable efficient parameter usage, but also achieve general-purpose computing power, outperforming other methods in numerous learning problems.

One limitation of this work is the number of additional hyperparameters, which prevents us from fully tuning Neurocoder. Our research aims to add new capabilities to current neural networks to improve their performance and make them robust in different learning scenario. Hence, we do not see any intermediate negative societal impact. In future work, we will extend Neurocoder's application beyond feed-forward layers. It would be interesting to efficiently replace all neural layers including CNN or Transformer by Neurocoder's programs. We can also further extend Neurocoder's ability by allowing a growing Program Memory, in which the model decides to add or erase memory slots as the number of data patterns grows or shrinks beyond the current program space's capacity. Such a system represents a more flexible general-purpose computer that can dynamically allocate computing resources by itself without human pre-specification.

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
