

Figure 4: Overview structure of Neurocoder. Main Network processes inputs to produce outputs. Program Memory stores modular units. Program Controller reads modular units from the Program Memory, composing Neural Programs for the Main Network in a data-driven manner. Each Neural Program is designed for specific input. Intuitively, it is analogous to building Lego structures corresponding to inputs from basic Lego bricks.

# Appendix

## Instance-based learning experiments

*Image classification-linear Main Network*    We used the standard training and testing set of MNIST dataset. To train the models, we used the standard SGD with a batch size of 32. Each MNIST image was flattened to a 768-dimensional vector, which requires a linear classifier of $7,680$ parameters to categorise the inputs into 10 classes. For Neurocoder, we used Program Memory with $P = 6$ and $K = 2$. The Program Controller's composition network was an LSTM with a hidden size of 8. We controlled the number of parameters of Neurocoder, which included parameters for the Program Memory and the Program Controller by reducing the input dimension using random projection $z_t = x_t U$ with $U \in \mathbb{R}^{768 \times 200}$ initialised randomly and fixed during the training. We also excluded the program integration to eliminate the effect of the residual program $\mathbf{R}$. Given the flattened image input $x_t$, Neurocoder generated the active program $\mathbf{P}_t$, predicting the class of the input as $y_t = argmax\,(x_t \mathbf{P}_t)$. The performance of the linear classifier was imported from [24] and confirmed by our own implementation.

*Image classification-deep Main Network*    We used the standard training and testing sets of CIFAR datasets. For most experiments, we use Adam optimiser with a batch size of 128. *The deep Main Networks were adopted from the original papers, resulting in 3-layer MLP, 5-layer LeNet [24] and* 100-*layer DenseNet* [18][1]. The other baselines for this task included a recent sparse Mixture of Experts (MOE [35]) and the Neural Stored-program Memory (NSM [22]). For this case, we employed the program integration with the residual program $\mathbf{R}$ to flexibly fit to the data distribution.

## Sequential learning experiments

*Synthetic polynomial auto-regression*    A sequence was divided into $n_{pa}$ chunks, each of which associated with a randomly generated polynomial. The degree and coefficients of each polynomial were sampled from $U \sim [2, 10]$ and $U \sim [-1, 1]$, respectively. Each sequence started from $x_1 = -5$ and ended with $x_T = 5$, equally divided into $n_{pa}$ chunks. Each chunk contained several consecutive points $(x_t, y_t)$ from the corresponding polynomial, representing a local transformation from the input to the output. Given previous points $(x_{<t}, y_{<t})$ and the current $x$-coordinate $x_t$, the task was to predict the current $y$-coordinate $y_t$. To be specific, at each timestep, the Main Network GRU was fed

---

[1]Only for experiments with DenseNet, to closely match the reported results, we followed the original training with SGD optimizer, scheduling learning rate and batch size of 32.

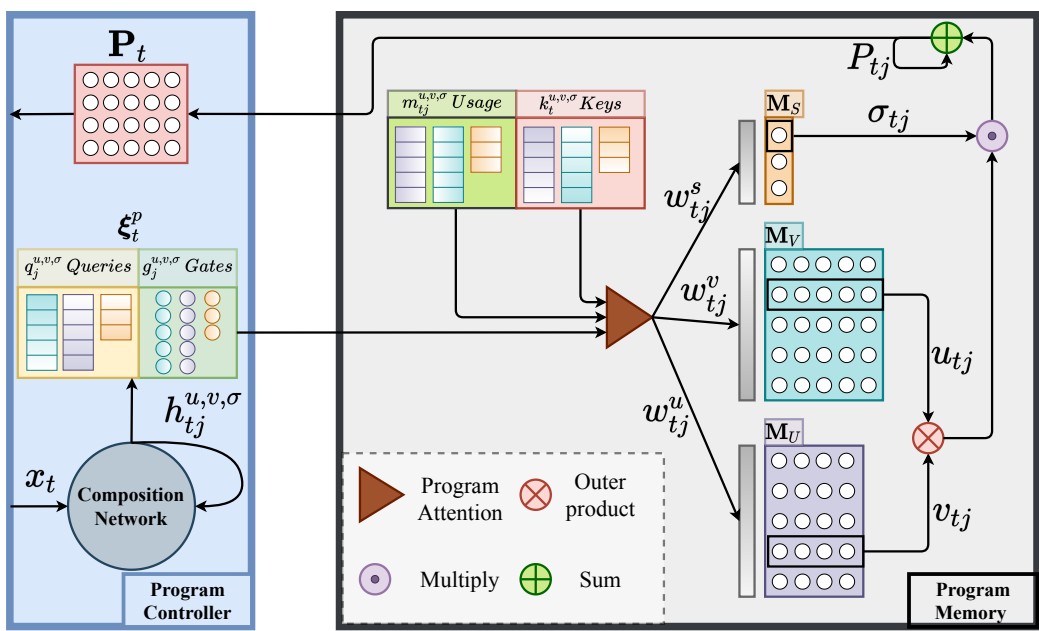

Figure 5: Active program coding. The Program Controller uses the composition network (a recurrent neural network) to process the input $x_t$ and generate composition signal $\boldsymbol{\xi}_t^p$, which is composed of the queries ($q$) and the interpolating gates ($g$). The similarity of the query to program memory keys ($k$) is then computed together with the memory usage ($m$) from which attention weights for the Program Memory are derived. The active program $\mathbf{P}_t$ is then "coded" through low-rank approximation using the $j$-th component accessed by recurrent attentions. For simplicity, one attention head is shown ($H = 1$).

with $(x_t, y_{t-1})$ and trained to predict $y_t$ by minimizing the mean square error $1/T \sum_{t=1}^{T} (\hat{y}_t - y_t)^2$ where $y_0 = 0$, $\hat{y}_t$ is the prediction of the network and $y_t$ the ground truth.

We augmented *GRU by applying Neurocoder* and HyperNet [13] *to the output layer of the GRU*. Here, the HyperNet baseline [13] generated adaptive scales for the output weight while the FiLM baseline [28] modulates the activation of the output layer. We trained the networks with Adam optimiser with a batch size of 128. To balance the model size, we used GRU's hidden size of 32, 28, 32, 16 and 8 for the original Main Network, HyperNet, FiLM, single-step and multi-step Neurocoder, respectively. We also excluded program integration phase in Neurocoders to keep the model size equivalent to or smaller than that of the Main Network.

We compared three configurations of Neurocoder - single-step, multi-head ($J = 1$, $H = 15$), multi-step, single-head ($J = 5$, $H = 1$) and multi-step without usage-based attention- against the original GRU with output layer made by MLP, HyperNet and FILM. We found that MLP failed to learn and converge within $10,000$ learning iterations. In contrast, both Neurocoders learn and converge, in as little as only $2,000$ iterations with the multi-step Neurocoder. HyperNet and FiLM converged much slower than Neurocoders and could not minimize the predictive error as well as Neurocoders when Gaussian noise (mean 0, variance $0.3 \times \max_t y_t$) is added or the number of polynomials ($n_{pa}$) is doubled (see Fig. 6).

***Atari 2600 games***   We used OpenAI's Gym environments to simulate Atari games. We used the standard environment settings, employing no-frame-skip versions of the games. The picture of the game snapshot was preprocessed by CNNs and the A3C agent was adopted from the original paper with default hyper-parameters as in [26]. *The actor/critic network of A3C was LSTM whose output layer's working program was provided by Neurocoder* or HyperNet. The hidden size of the LSTM was 512 for all baselines.

*Seaquest and MsPacman*. The original A3C agent was able to learn and obtain a moderate score of around $2,500$ after 32 million environment steps. We also equipped A3C with HyperNet-based ac-

tors/critics, however, the performance remained unchanged, with scores of about $2/3$ of Neurocoder-based agent's.

**Multi-task learning experiments**

Table 4 lists the input-output structure and configuration of the algorithmic tasks: Copy, Repeat Copy, Associative Recall and Priority Sort. Basically, each timestep of the input sequence presents a binary vector of 8 bits. In our multi-task setting, the input-output structure is $t_1, x^1_{1...I_1}, t_2, x^2_{1...I_2}, t_3, x^3_{1...I_3}, t_4, x^4_{1...I_4} \rightarrow y^1_{1...O_1}, y^2_{1...O_2}, y^3_{1...O_3}, y^4_{1...O_4}$ where $\left\{ t_i, x^i_{1...I_i}, y^i_{1...O_i} \right\}^4_{i=1}$ is the task identification, input and output sequence of the task, respectively. The multi-task learning problem was challenging because the models must distinguish tasks by remembering the task identifications and learn to solve different algorithms by generating different interface programs in accordance with each task.

Following [22], we used the same NTM with 1 read and 1 write head, and applied Neurocoder and other conditional computing methods to the interface network of the NTM, which is a single-layer MLP with $\tanh$ activation. We used single-step attention Neurocoder for this task to keep the number of parameters comparable with other models. We trained the models with RMSProp optimiser with learning rate of $10^{-4}$ and batch size of 64 to minimise the cross-entropy loss of the ground truth output and the predicted one. The evaluation metric was $\%$ bit error, which was computed for a sequence as $\frac{\# \text{ wrong bits}}{\# \text{ total bits}} \times 100$.

**Continual learning experiments**

*Split MNIST*   We used the same 2-layer MLP and continual learning baselines as in [17]. Here, we again excluded program integration to avoid catastrophic forgetting happening on the residual program $\mathbf{R}$. Remarkably, the NSM with much more parameters could not improve MLP's performance, illustrating that simple modular conditional computation is not enough for continual learning (see Table 5).

*Split CIFAR*   The 18-layer *ResNet* implementation was adopted from Pytorch's official release whose weights was pretrained with ImageNet dataset. When performing continual learning with CIFAR images, we froze all except for the output layers of ResNet, which was a 3-layer MLP. We only tuned the hyper-parameters of SI and Neurocoder for this task.

In the CIFAR10 task, compared to the monolithic ResNet, the Neurocoder-augmented ResNet could achieve much higher accuracy when we finished the learning for all 5 tasks ($55\%$ versus $70\%$, respectively). Also, we realised that stressing the orthogonal loss further improved the performance. When we employed Synaptic Intelligence (SI [41]), the performance of ResNet improved, yet it still dropped gradually to just above $70\%$. In contrast, the Neurocoder-augmented ResNet with SI maintained a stable performance above $80\%$ accuracy (see Fig. 8 (left)).

In the CIFAR100 task, Neurocoder alone with a bigger program memory slightly exceeded the performance of SI, which was about $10\%$ better than ResNet. Moreover, Neurocoder plus SI outperformed using only SI by another $10\%$ of accuracy as the number of seen tasks grew to 20 (see Fig. 8 (right)).

**Training procedure and hyper-parameter selections**

For all experiments, Neurocoder was jointly trained with Main Networks. We trained all the models using single GPU NVIDIA V100-SXM2. Running time depends on task, the longest task is multi-task learning with MN as NTM, which took 1 day for 1 training run with Neurocoder. Adding Neurocoder makes the training slower about 30%, yet still faster than MOE or NSM. However, compared to HyperNet or FILM, Neurocoder is still slower by 15%, which is the limitation of Neurocoder.

The learning rate of optimisers was set to default value unless stated otherwise. The Main Network's hyper-parameters were fixed and we only tuned the hyper-parameters of Neurocoder and its competitors: MOE, NSM, HyperNet and FILM. In particular, for Neurcocoder, main hyper-parameters such as number of memory slots ($P$), recurrent steps ($J$), and heads ($H$) were selected from $\{10, 20, 30, 50, 80, 100\}$, $\{1, 5\}$ and $\{1, 5, 15\}$, respectively. Hyper-parameters such as num-

| Notation | Meaning | Location | |
|---|---|---|---|
| | | Program Controller | Program Memory |
| Trainable parameters | | | |
| $\theta^{u,v,\sigma}$ | Composition network | ✓ | |
| $\phi$ | Integration network | ✓ | |
| $\varphi^{u,v,\sigma}$ | Key generator network | | ✓ |
| $\mathbf{R}$ | Residual program (optional) | ✓ | |
| $\mathbf{M}_U$ | Memory of left singular vectors | | ✓ |
| $\mathbf{M}_V$ | Memory of right singular vectors | | ✓ |
| $\mathbf{M}_S$ | Memory of singular values | | ✓ |
| Control variables | | | |
| $\boldsymbol{\xi}_t^p$ | Composition control signal | ✓ | |
| $\boldsymbol{\lambda}_t^p$ | Integration control signal | ✓ | |
| $k^{u,v,\sigma}$ | Program keys | | ✓ |
| $m^{u,v,\sigma}$ | Program usages | | ✓ |
| Hyper-parameters | | | |
| $P$ | Number of memory slots | | ✓ |
| $K$ | Key dimension | | ✓ |
| $l_I$ | Number of considered least-used slots | | ✓ |
| $J$ | Number of recurrent attention steps | ✓ | |
| $H$ | Number of attention heads | ✓ | |
| $a$ | Orthogonal loss weight | | ✓ |

Table 2: Important parameters of Neurocoder.

| Architecture | Task | Original | MOE | NSM | Neurocoder |
|---|---|---|---|---|---|
| MLP | CIFAR10 | 52.06 | 50.76 | 52.76 | **54.86** |
| | CIFAR100 | 23.31 | 22.79 | 25.65 | **26.24** |
| LeNet | CIFAR10 | 75.71 | 75.88 | 75.45 | **78.92** |
| | CIFAR100 | 42.73 | 42.47 | 43.14 | **47.21** |
| DenseNet | CIFAR10 | 93.61 | 80.61 | 94.24 | **95.61** |
| | CIFAR100 | 78.11 | 69.48 | 71.76 | **79.34** |

Table 3: Best test accuracy over 5 runs on image classification tasks comparing original architecture, Mixture of Experts (MOE), Neural Stored-program Memory (NSM) and our architecture (Neurocoder). Three architectures of the Main Network of Neurocoder were considered: 3-layer perceptron (MLP), 5-layer CNN (LeNet [24]) and very deep Densely Connected Convolutional Networks (DenseNet [18]). We employed two classical image classification datasets: CIFAR10 and CIFAR100.

ber of least-used slots ($l_I$) key dimension ($K$), orthogonal loss weight ($a$) was selected from $\{2,5\}$, $\{3,5\}$ and $\{0.1,10\}$, respectively.

For MOE, we tuned the total number of experts and top-$k$ chosen experts from range $\{10,50,80,100\}$ and $\{1,5,10\}$, respectively. For NSM, we tuned the number of program memory slots $\{5,10,50\}$. Other hyper-parameters of MOE and NSM were kept as in the original papers. For HyperNet and FiLM, chosen as MLPs (ReLU activation), we tuned the number of layers $\{1,2\}$ and hidden size $\{64,128,256\}$.

We report details of best hyper-parameters and model size for each tasks in Table 6 and 7, respectively. Readers are referred to Table 2 for the complete list of parameters in Neurocoder.

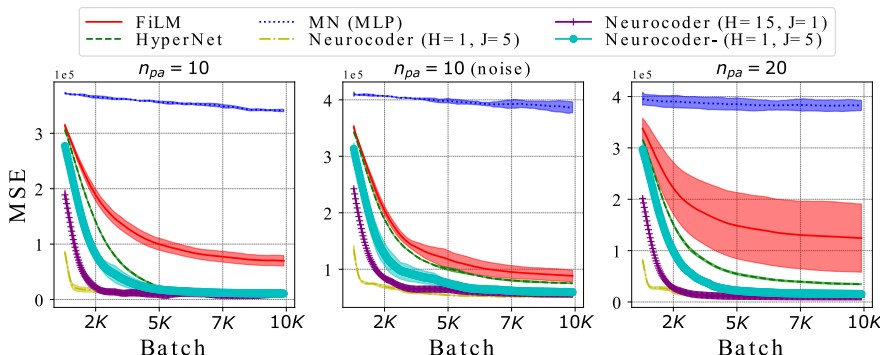

Figure 6: Polynomial auto-regression: mean square error (MSE) over training iterations with a batch size of 128 comparing FiLM, HyperNet, Main Network (MLP), single-step, multi-step Neurocoders. - denotes the ablated Neurocoder without usage-based attention. The learning curves are taken average over 5 runs.

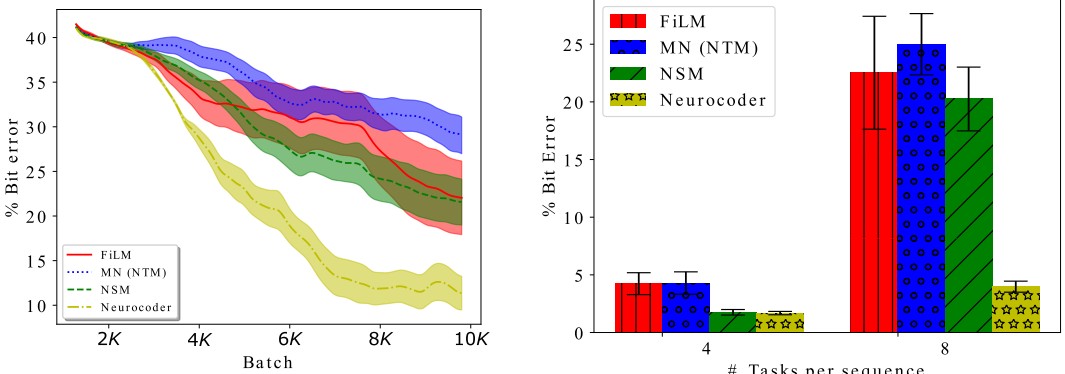

Figure 7: Multi-algorithm learning task (mean and std. over 5 runs). Left: Bit error over training steps (8 tasks per sequences). Right: Average best bit error on different testing settings. Lower is better.

| Tasks | Configuration | Input → Output |
|---|---|---|
| Copy | Sequence length range: [1, 3] | $x_1, ..., x_T \rightarrow x_1, ..., x_T$ |
| Repeat Copy | Sequence length range: [1, 3] 
 #Repeat range $n$: [1, 2] | $n, x_1, ..., x_T \rightarrow [x_1, ..., x_T] \times n$ |
| Associative Recall | #Item range: [2, 3] 
 Item length: 2 | $[x_{1,1}, x_{1,2}], ..., [x_{T,1}, x_{T,2}], [x_{i,1}, x_{i,2}] \rightarrow [x_{i+1,1}, x_{i+1,2}]$ |
| Priority Sort | #Item: 3 
 #Sorted Item: 2 | $[x_1, p_1], [x_2, p_2], [x_3, p_3] \rightarrow x_{i_1}, x_{i_2}$ s.t. $p_{i_1} \geq p_{i_2} \geq p_{i_3}$ |

Table 4: Algorithmic tasks used in multi-task learning.

| Method | MN (MLP [17]) | MN (MLP ours) | NSM | Neurocoder |
|---|---|---|---|---|
| Adam | 93.46±2.01 | 93.75±3.28 | 87.55± 4.38 | **96.54±1.39** |
| Adagrad | 98.06±0.53 | 98.02±0.89 | 96.63±1.49 | **99.01±0.19** |
| L2 | 98.18±0.96 | 98.14±0.43 | 91.44± 3.80 | **98.35±0.74** |
| SI | 98.56±0.49 | 98.69±0.20 | 98.87±0.20 | **99.14±0.24** |
| EWC | 97.70±0.81 | 97.00±1.10 | 93.94±2.36 | **97.88±0.22** |
| O-EWC | 98.04±1.10 | 98.23±1.17 | 96.11±1.27 | **98.30±1.48** |

Table 5: Incremental task continual learning with Split MNIST. Final test accuracy (mean and std.) over 10 runs.

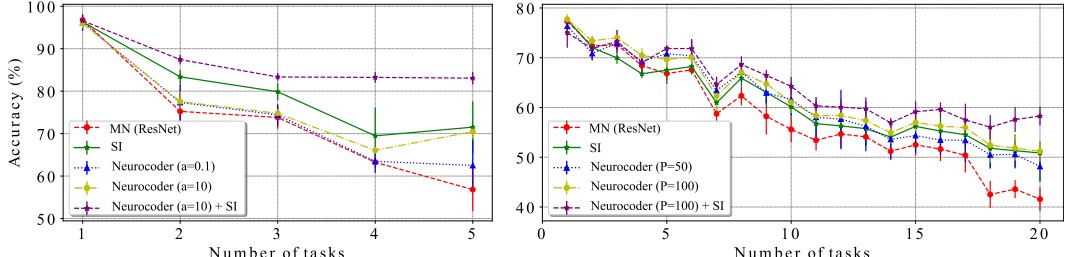

Figure 8: Incremental task continual learning with Split CIFAR10 (left) and CIFAR100 (right). Average classification accuracy with error bar over all learned tasks as a function of number of tasks.

| Task | Neurocoder | Use **R** |
|---|---|---|
| MNIST | $P = 5, J = 5, H = 1$
$K = 2, l_I = 2, a = 0.1$ | X |
| CIFARs | $P = 30, J = 5, H = 3$
$K = 5, l_I = 5, a = 0.1$ | ✓ |
| Polynomial
auto-regression | $P = 10, J = 1, H = 15 \quad P = 20, J = 5, H = 1$
$K = 3, l_I = 0, a = 0.1 \quad K = 3, l_I = 2, a = 0.1$ | ✓ |
| Atari games | $P = 80, J = 1, H = 15, K = 3, l_I = 5, a = 0.1$ | ✓ |
| Multi-algorithm | $P = 30, J = 1, H = 5, K = 5, l_I = 2, a = 10$ | ✓ |
| Split MNIST | $P = 50, J = 1, H = 10, K = 5, l_I = 5, a = 10$ | X |
| Split CIFARs | $P = 100, J = 1, H = 10, K = 5, l_I = 5, a = 10$ | X |

Table 6: Best hyper-parameters of Neurocoder in all experiments. For polynomial auto-regression task, two Neurocoder configurations are included, corresponding to single-step and multi-step Neurocoder.

| Task | Main Network | Original | MOE | NSM | HyperNet | FiLM | Neurocoder |
|---|---|---|---|---|---|---|---|
| MNIST | Linear classifier | 7.8K | – | – | – | – | 7.3K |
| | 3-layer MLP | 1.7M | 15.4M | 21.2M | – | – | 1.9M |
| CIFARs | LeNet | 2.1M | 12.3M | 27.1M | – | – | 2.3M |
| | DenseNet | 7.0M | 20.5M | 16.7M | – | – | 7.3M |
| Polynomial
auto-regression | GRU | 3.4K | – | – | 3.5K | 3.6K | 3.6K  2.1K |
| Atari games | LSTM | 3.2M | – | – | 3.6M | – | 3.3M |
| Multi-algorithm | NTM | 308K | – | 264K | – | 254K | 255K |
| Split MNIST | 2-layer MLP | 328K | – | 2.3M | – | – | 348K |
| Split CIFARs | ResNet | 12.6M | – | – | – | – | 12.6M |

Table 7: Number of parameters of machine learning models in all experiments. The parameter count includes the parameter of the Main Network and the conditional computing model. – denotes not available. For tasks that contain different datasets, leading to slightly different model size, the numbers of parameters are averaged. For polynomial auto-regression task, two Neurocoder configurations are included, corresponding to single-step and multi-step Neurocoder.