# OpenReview forum: "Neurocoder: Learning General-Purpose Computation Using Stored Neural Programs"
_NeurIPS.cc/2021/Conference — NeurIPS 2021 Submitted_

### Official Review · Reviewer_w4yd · 2021-07-12

**Rating:** 7
**Confidence:** 3

**Summary:**

This paper proposes Neurocoder, a new general purpose neural network that “codes” itself in a data responsive way by composing relevant programs from a set of shareable, modular programs stored in external memory. Unlike previous work, Neurocoder treats modular programs as sharable datum in memory and composes relevant programs from stored “programs”. Experimental results show performance improvement in solving tasks such as object recognition, playing video games, and continual learning.



**Limitations And Societal Impact:**

To me, this work does not have negative societal impacts (including potential malicious or unintended uses, environmental impact, fairness considerations, privacy considerations, and security considerations).

**Main Review:**

The proposed framework Neurocoder is interesting. The idea of learning and storing programs as external memory is interesting.
To demonstrate the flexibility of Neurocoder framework, the authors consider different learning paradigms: instance-based, sequential, multi-task and continual learning. The experiments were well conducted and demonstrated the effectiveness of the proposed framework. The proposed framework also required a much smaller number of parameters.

The general idea of Neurocoder is to retrieve relevant programs from memory using the attention mechanism. Similar ideas have been proposed in previous work such as the end-to-end memory networks (Sukhbaatar et al.2015), the neural turing machine (Graves et al. 2014), and the neural module networks (Jacob et al. 2016). It would be better if the authors explicitly describe the similarities and differences between Neurocoder and the previous work, and their novel contributions.

Neurocoder induces modular programs and store them in external memory, so scalability may become an issue, which needs to be evaluated as well. In other words, how many programs can Neurocoder store in its external memory?

The proposed approach treats programs in the form of matrices/neural networks. There are some related work that synthesizes actual programs through input/output. For example:

Shu et al., Neural Programming by Example, Proc. the 31st AAAI Conference on Artificial Intelligence (AAAI 2017), February 2017, San Francisco, California, USA.

Minor:

. In Figure1, why only the last layer of the main network is adaptively loaded?


**Time Spent Reviewing:**

3 hours

---

> ### Author Response · Authors · 2021-08-09
> **Reply to w4yd**
>
> We thank the reviewer for your insightful and constructive comments. We address your concerns in the following.
>
> $\textbf{"It would be better if the authors explicitly ..."}$ We agree and will revise our paper accordingly.
>
> $\textbf{"How many programs can Neurocoder store in its external memory?"}$ In Sec. 3.4, we tested up to 100 programs. Since our programs are singular programs (vectors), our program memory can scale normally as other data memory in normal memory-augmented neural networks. Unlike others (NSM or MoE) wherein scaling the program memory requires huge physical memory and computation budget, our does not.
>
> $\textbf{"There are some related work ..."}$ We agree and will add references to the program synthesis approach. Thank you for your suggestion.
>
> $\textbf{Fig. 1}$ For ease of visualization, we only depict the case P is used for the last layer, which is the common setting in our experiment. Yet technically, any layer can be loaded with P. We also have experiments where all layers of MLP are loaded (e.g. Sec 3.1 and 3.4)

---

### Official Review · Reviewer_DMv6 · 2021-07-15

**Rating:** 7
**Confidence:** 5

**Summary:**

This paper approaches the problem of multi-purpose neural networks that can learn several tasks or several methods for computing a task with a unique approach. This approach involves representing the last portion of a neural network as a task-configurable set of weights. The way this is accomplished is by storing several neural networks in memory in a compressed format, utilizing SVD, and then using the input to produce attention weights on each of the S V and U components of the SVD, thus reconstructing the weights, adding in a non-task-specific residue representing the remainder of the weights, and then using those weights for the remainder of the network.

The paper then uses several standard benchmarks both in standard learning tasks, RL tasks, and continual learning tasks to demonstrate that their network is able to outperform similarly powerful networks that do not contain the specific properties that they have built into their network.

**Limitations And Societal Impact:**

The authors seem aware of the limitations of their work and present them clearly. I agree with the authors that there are probably not many direct implications for this work, though I would say that there are possibly some positive implications if this technique in the long run allows for more efficient training of neural networks.


**Main Review:**

Originality: the idea of attending directly over a compressed representation of a neural networks’ weights appears to me to be novel, as is the presented architecture in general.

Quality: the paper presents a very interesting concept and has the experiments to demonstrate it in detail. I especially like the way that the compressed format is used to save on the storage efficiency of the program, and I wonder if it might also have some regularization advantages.

The one major critique I might give is that the idea seems to have a lot of components (the weights on the residual program, the use of the usage counters to promote less used programs, the idea that the singular values need be ordered, etc.) whose importance is only explained using theoretical justifications and some ablation analysis might go some way as to justifying the need for so many parts.

Clarity: While the high level of this paper seems really interesting, the clarity of the paper is lacking in a number of areas, to the point that I am having a hard time understanding many of the details being described, even after reading through the paper a number of times. I think this submission would be improved dramatically by addressing some of these concerns.

- The section “neurocoder stores singular value decomposition of neural programs in program memory” I think could use much more direct information about the sizes of the various tensors involved. For example, even after reading through the paper twice I am unclear on what exactly P is: is it a matrix where rows are programs and columns are the parameters in each program? Are the matrices ones that represent the parameters in one program (somehow arranged into 2d) and there are several such matrices making P a tensor? Is it something else entirely? I lean towards the second explanation because that’s what M_U, M_V, M_S implies to me, but I am not sure. Additionally, I think equations that are fully qualified with all the relevant symbols would also be helpful in understanding exactly what is going on, even if it makes the equations somewhat less easy to read at first glance.
- When you say “the singular programs are trained to represent unitary functions necessary for any computation whilst the composition and integration networks are trained to compose the relevant programs for the considering task” I believe that you are describing the result of an end-to-end training process rather than these networks being trained individually, but I am not 100% sure. The next sentence beginning “as such” seems disconnected from this point as well as it describes an auxiliary loss that seems to enforce the structure of the singular programs rather than anything to do with their function.
- This might be an artifact of a misunderstanding created by the previous point, but I do not understand what’s going on in equations 3 and 4. Are those just linear combinations of attention weights and the memory modules for each of U and V? If so, why are the outputs denoted as vectors? Or since we’re retrieving the nth vector, why is n not one of the indices in M_U or M_V (both of which seem to be conditioned only on i).
- The section “recurrent access to the program memory via the composition network” confused me at first because I thought you were accessing the program memory recurrently, that is feeding back information about the program memory into the recurrent module in order to inform future key generations. Based on a closer reading however, it seems that what you are doing is generating several queries that are dependent on the previous *queries* but not dependent on the state of the program memory in any way. If this is accurate, I think the wording should be updated to make it more clear. Perhaps “rolling out query heads rather than generating them in parallel allows us to ensure that each of the query heads is distinct from the rest and thus allows us to avoid program collapse”
- Equation 12: I feel  like I’m missing something about how you are suppressing the magnitude of R, surely since R is a purely trainable parameter it can grow to an arbitrary size to counteract any suppression that might occur?
- Figure 1: the diagram suggests that the residual program is generated out of the input separately from the composition network. I believe that one of the weights from the composition network is used here so an arrow should probably be added from the composition network to the residual program.

Significance: the paper seems to be fairly significant and I can see this approach being directly used with pretty much any architecture that seeks to solve multiple tasks or a single but varied task. I also see extensions of this work being applied in a number of different domains.


**Time Spent Reviewing:**

2

---

> ### Author Response · Authors · 2021-08-09
> **Repy to Reviewer DMv6**
>
> We thank the reviewer for your valuable comments. In the following, we address your concerns.
>
> $\textbf{"The section "neurocoder stores ..."}$ As stated in L65, P is an active program, representing a neural program loaded to the neural network. For simplicity, we define a neural program as a weight of the neural network, so P is always a matrix (2d tensor). We can think of it as the parameters of one program corresponding to one layer of the neural network. If the network has multiple layers, we have multiple Ps. We will revise the section to make this point clearer.
>
> $\textbf{"When you say ..."}$ Yes, it is trained end-to-end. The auxiliary loss is to ensure singular programs are orthogonal, which implies the functions represented by them are unitary (not overlapping).
>
> $\textbf{"This might be an artifact ..."}$ There seems to be a misunderstanding. Eq. 3 and 4 are linear combinations of attention weights and the rows of M_U and M_V. M_U(i) denotes the i-th row of M_u, so the output is a vector. n is the index of a singular component, which is not necessarily related to i. For example, in an extreme case, we can retrieve 100 vectors (n=1 to 100) from a M_U of 5 rows (i=1 to 5). Also, we are performing soft attention and your suggestion seems to be hard attention, which is trickier to train.
>
> $\textbf{"The section “recurrent access ... "}$ Your comment is correct. We will revise our paper to make it clearer based on your suggestion.
>
> $\textbf{Eq. 12}$ Yes, R can grow to encounter the suppression. Our simple method may not be a strict solution. Yet, it prevents R from freely growing. At least, the gradient backpropagated to R will be suppressed and slow down the learning of R. When the weight gets close to zero, the R contribution is almost ignored.
>
> $\textbf{Figure 1}$ You are right. We missed $\sigma_{tr_{m}}$ from the composition network. We will add this to Fig. 1. Thank you for pointing it out.

---

> > ### Comment · Reviewer_DMv6 · 2021-08-23
> > **Clarity improvements appreciated but ablation analysis would be ideal**
> >
> > Thank you for your responses to my and other reviews regarding clarity. I think with these changes the paper would be easier to read.
> >
> > I have increased my score to a 7. However, I do think it might be helpful to provide some ablation analysis in this paper.

---

> > > ### Author Response · Authors · 2021-08-24
> > > **Thank you**
> > >
> > > Thank you for increasing your score. In the revision, we will make our ablation study clearer in the experiment section.

---

### Official Review · Reviewer_4P5s · 2021-07-16

**Rating:** 7
**Confidence:** 2

**Summary:**

This paper presents an architecture where the weights $P$ of a neural network’s linear layer can be determined by the input. The weight matrix $P$ of the linear layer is constructed by using its singular value decomposition $P=SEV$, with each of the columns of $S$ and $V$ determined by attending over a set of ``singular programs’’. Authors also use recurrent multi-headed attention to select multiple singular programs, using both content-based attention but also usage-based attention to improve the rank of $P$.

The authors run a range of experiments to test the architecture in a variety of settings. Starting with instance-based learning with MNIST/CIFAR, the author justifies the use of recurrent attention, and visualizes the attention pattern. The authors further justifies the use of recurrent attention with a piecewise polynomial regression sequential learning experiment, and demonstrates the attention shift at the polynomial boundaries. In the reinforcement setting, authors shows that their approach enables A3C to learn in an environment with sparse rewards. Finally, the authors demonstrate that the approach generalizes well in a multi-task learning setting, and that the approach mitigates catastrophic forgetting issues in a continual learning setting.


**Limitations And Societal Impact:**

I can't think of positive/negative societal impact beyond using the typical neural network architectures.

**Main Review:**

I found the central idea in the paper elegant and compelling: construct the weights of a MLP layer using its singular-value decomposition, where the singular vectors are selected (using attention) from a library of these vectors (aka ``singular programs’’). The use of recurrent attention is well justified with the experiments. The potentially low rank of the active program is fixed using a residual program.

One part of the architecture that I did not find compelling is the selection of the singular values using attention. It seemed odd to store and select these *scalar* values using attention, rather than computing the singular values directly using the attention weights (i.e. using $w_{in}^\sigma$ in equation 5, without using $M_s(i)$ at all). I’m also not sure why the enforcement of the order of singular values is necessary: i.e. why not directly generate the singular values along with the attention weights using the recurrent attention mechanism?

The selection of experiments are impressive, although it is unfortunate that most of the details are in the supplementary material and are challenging to find. The details I had to hunt for include: (1) the authors mentioned in the MNIST experiment that the number of parameters of the baseline vs neural coder are comparable, and I did not find the comparison, (2) the author mentions that ablation studies are done, and it would have been helpful to see this information consolidated somewhere.

The writing is understandable, although it wasn’t until I saw the MNIST experiment hyperparameters that I understood which parts of the model were the trainable weights (which might just be me).

Here are some more detailed feedback:


Abstract line 8-9: I don’t know if the claim that Neural Program is treated as datum in memory is a fair statement, given the wealth of prior work that the authors cite.

Appendix Figure 4: I recommend against using this analogy and omitting this figure. Experienced readers would not need this figure, and that inexperienced readers will draw misconceptions from this figure. The figure makes it look like the model is trying to reconstruct the input. It also made me think that the program memory was fixed, rather than a trainable weight.

Figure 1: Minor suggestion to make $M_u$ a $6x4$ matrix and $M_v$ a $6x5$ matrix to show that you’re storing some number of singular vector (and that this number is a hyperparameter independent of the number of columns in $P$). Additionally, it is strange that $M_s$ has fewer columns than $M_u$ and $M_v$

If you need to cut content, the last two sentences in lines 82-84 can be cut. Additionally, the introduction can be tightened up significantly.

Figure 2(b) Are you showing the first *3 singular vectors* or the first *5*? Looks like the x-axis in the plot ranges over 5 values. I’m also curious as to why, in the last column, the last two singular vectors look virtually identical for all the digits.

Figure 2(d) Maybe helpful to make the x-axis range consistent. It is strange that even though the polynomial does not change in some transitions (e.g. x=200, 250, 300 in the top figure), the attention weights sometimes do (e.g. x=250, but not x=200) Conversely, there is a change in the polynomial in the transition x=350, but the attention weight stays the same.


**Time Spent Reviewing:**

7

---

> ### Author Response · Authors · 2021-08-09
> **Reply to Reviewer 4P5s**
>
> We thank the reviewer for your insightful and constructive comments. We address your concerns in the following.
>
> $\textbf{"One part of the architecture ..."}$ Computing directly will not utilize the stored values that are updated through backpropagation. Our mechanism is more general with $\mathbf{M_S}(i)$. For example, it becomes your suggestion if $\mathbf{M_S}\left(i\right)$ is learnt to be 1. We need to enforce the order to simulate the SVD (the first component has the highest singular value and hence is the most important). This gives some meaning for each attention and hopefully, the model can learn to reflect that meaning (e.g. the first attention is to construct the most important component of the program and so on).
>
> $\textbf{"(1) the authors mentioned ... "}$ We mentioned it in the main text (L294) and asked readers to refer to Appendix Table 7 for that information.
>
> $\textbf{"(2) the author mentions that ..."}$ We do not have a separate section for ablation study. Rather, we embed the ablation into the tasks. For example, in the object recognition task, we examine different recurrent attention steps (Fig. 2); in the polynomial auto-regression task, we ablate the usage-based attention (L233); in the continual learning task, we examine a different number of programs and the impact of the orthogonal loss (L283).
>
> $\textbf{L8-9}$ We made a claim of $\textit{efficient}$ treatment of neural program as a datum. Other works store the whole weight as a datum, which is inefficient.
>
> $\textbf{Appendix Fig. 4}$ We agree and will remove it. Thank you for your suggestion.
>
> $\textbf{Fig. 1}$ We agree with your suggestions. For $\mathbf{M_S}$, we believe you meant the number of rows since singular values are scalar (#column=1). Note that the numbers of rows of $\mathbf{M_U}$, $\mathbf{M_V}$ and $\mathbf{M_S}$ are not necessarily equal (Line 121-122). Yet, to reduce the number of hyperparameters, we set them to be equal.
>
> $\textbf{Fig. 2(b)}$ We showed the first 5 components (we will fix the typo in the caption). We note that for digit 7, the last two patterns are different and for other digits, they are identical. The phenomenon may reflect the order of importance of the attention we explained above. Later singular vectors represent less important components. Most of the digits can be discriminated by early attentions, so the later ones can be similar (the singular values for later components can be too small, suppressing any gradient updates through later attention).
>
> $\textbf{Fig. 2(d)}$ For x=200,250,300, we note that the range of the y-axis is big. If we zoom in, we can see better the change in the function. For x=350, if we look closely, the weight patterns change (rows 4-6). But we agree that the change is small compared to that of the functions. Hence, the behaviour is imperfect and there is room for improvement.

---

### Official Review · Reviewer_D8Dy · 2021-07-17

**Rating:** 6
**Confidence:** 4

**Summary:**

This paper proposes a general-purpose neural architecture with an explicit goal of modularity, called Neurocoder. Neurocoder dynamically produces the weights for a layer in a neural network, based on the input, by combining "programs" from memory. These programs consist of left and right singular vectors ($U$, $V$) and singular values ($S$). For a given input, a recurrent multi-head process is used to retrieve the program from memory. In each step of the recurrent process, we obtain a query $q$ and gate $g$ for each of the $H$ heads and for each of the $U$, $V$ and $S$. The query $q$ is used for content-based attention inside $U$, $V$, and $S$ respectively. The gate $g$ is used to combine the content-attention result with a least-recently-used attention. By putting all of this together we obtain a $u_n$, $v_n$, and $\sigma_n$ for each head in each step of the recurrent process. We then sum $\sum_n \sigma_n u_n v_n^T$ to obtain the weights for the layer.

The authors evaluate this architecture on MNIST, CIFAR, RL on Atari games, and polynomial regression.

**Limitations And Societal Impact:**

I think the biggest limitation of the work is the lack of ablation experiments, especially considering the complexity of the approach. The paper states on line 184 that "For some experiments, we include ablation studies" but I was unable to find any in the paper or appendix, except for varying the number of recurrent attention steps. Some example things that you can remove or adjust:
- Storing the programs in full-rank form rather than as $U$ and $V$
- Least-recently-used attention
- Heads
- Residual program

It would be particularly insightful if the paper can describe this work and the related prior work (HyperNetworks, FiLM, mixture of experts, Neural Stored-Program Memory) within some general framework where different axes of variation within the framework lead to the prior works and this one.

While not technically an ablation, I suggest the authors to also report what happens when the Neurocoder layer is inserted in different places inside a neural architecture.

**Main Review:**

## Originality
Overall the paper seems similar to the previous work Neural Stored-Program Memory, but with differences such as the program vectors being stored in three parts as the singular vectors $U$, $V$ and singular values $S$; incorporating least-recently-used attention; using recurrence to build the output weight matrix over several time steps; and integrating a "residual program" that is fixed for every input and added to the dynamically-computed weights.

## Quality
The paper aims to address important challenges about modularity and conditional computation in neural networks, and shows improvements upon baselines (although not SoTA results) on any task. Here are some general comments:
- It's not clear if the proposed architecture addresses catastrophic forgetting in the manner mentioned, "attending to different singular programs whilst learning different tasks". It would be interesting to see whether this is actually the case, or if different
- I felt that it is a bit strange to store and retrieve singular values from the memory, since they are scalars. Some statements in the paper don't make sense considering they are scalars, such as line 148-19 "$f_\varphi$ learns to compress each memory slot into a $K$-dimensional vector". I am not sure attention over scalars makes a lot of sense either, especially considering that the retrieved values are processed further to ensure monotonicity in equation 5.
- There is no mention of the computational complexity of the approach. It seems like it would be rather high considering the complexity of the approach, and lack of parallelization opportunities (for example, due to the recurrence).
- It would be useful to see what happens if the Neurocoder layer is used in different parts of the neural network, rather than just the output layer.

## Clarity
The overall model is quite complicated, and I would focus on making sure that the paper describes it clearly as possible. Some examples:
- The notation can clarify the type of each variable. For example, in equation 1, my understanding was that $\sigma_n$ is a scalar while $u_n$ and $v_n$ are (column) vectors, but there is no specific notation to clarify this.
- In line 112, $\mathcal{L}_{o}$ is a matrix, but it is added to a scalar in equation 2. It should be clarified how that matrix is turned into a scalar loss.
- I felt that it was confusing to write $w_{tijh}$ as there are too many subscripts. I would drop $t$ since it is not relevant, flatten $j$ and $h$ into $n$ and make it clear in the recurrent section how the heads and steps map to $n$, and distinguish $i$ since it is used for softmax.

Although it could take some space due to the complexity of the work, it would be very helpful as well to have the full set of equations for the computations performed written down together in one place.

I couldn't understand the binary decision tree in figure 2(c) and lines 202-207 and how that is supposed to relate to the attention patterns.

## Significance
The empirical results seem to improve upon the prior work and the work contains interesting ideas that others may build upon. However, the complexity of the approach will likely limit further adoption.

**Time Spent Reviewing:**

3

---

> ### Author Response · Authors · 2021-08-09
> **Repky to Reviewer D8Dy**
>
> We thank the reviewer for your valuable comments. In the following, we address your concerns.
>
> $\textbf{Quality}$
>
> $\textbf{"It's not clear if ..."}$ Thank you for your suggestion. We will have a visualization for this. We expect similar things to Fig. 2 (d).
>
> $\textbf{"I felt that it is a bit strange ..."}$ We perform attention to the key vectors. As long as the singular value is well presented by the key vectors (program name), there is no problem with this mechanism. We train $f_{\varphi}$ for mapping from the singular value to the key. For example, if the singular values are discrete, $f_{\varphi}$ can learn one-hot like representations of the singular values. Everything is trained end-to-end to maximize the final performance so we let $f_{\varphi}$ automatically determine the mapping. That said, we agree that for singular values, it is about representation, not compression. We will revise our sentences accordingly.
>
> $\textbf{"There is no mention of the computational complexity ... "}$ We mentioned it in Appendix L519-524. In short, Neurocoder was slower than FILM by 15% but much faster than sparse MoE or NSM. We will add more details in the revised paper.
>
> $\textbf{"It would be useful to see what happens if ... "}$ As stated in L189, we also used Neurocoder for all layers of MLP used in our experiments.
>
> $\textbf{Clarity}$
>
> $\textbf{L112}$ it is a typo. We missed the L2-norm symbol in defining $\mathcal{L}_{o}$.
>
> $\textbf{"I felt that it was confusing ..."}$ t is here to make it clear that the process is executed for each sample (conditional computation). j and h can be flattened to n (as explained L141), but later we need to mention them separately (Eq. 9, 10). We agree that it can be improved and we will further simplify this notation based on your comment.
>
> $\textbf{Fig 2c}$ In the caption, we described the graph. Here we provide more explanation. You can see there are only 2 patterns of attention (let call them UP and DOWN). When looking at the attention over time in Fig. 2(b), you can see the correspondence. From left to right, using the UP (DOWN) attention pattern corresponds to moving up (down) in the tree. Among 10 digits, using UP at the first attention step are {1,2,3,5,6,7}. The other uses DOWN {0,4,7,9}. These form the two nodes at the first depth of the tree (similar interpretation for other attention steps and deeper nodes in the tree).
>
> $\textbf{Limitation}$
>
> $\textbf{"Storing the programs in full-rank ..."}$ Your suggested baseline is similar to the NSM baseline we already used across experiments.
>
> $\textbf{Least-recently-used attention}$ We did an ablation study in Synthetic polynomial auto-regression task. See more in Appendix Fig. 6.
>
> $\textbf{Heads}$ We currently do not have an ablation study for this one. We find it depends on tasks. Some tasks may need more heads while others do not (Appendix Table 6 reports the best values). We will add this ablation study in the revised paper.
>
> $\textbf{Residual program}$ Again, it depends on tasks. For continual learning, having a residual program hurts performance since catastrophic forgetting will attack the residual program. For image classification, the residual program helps. We did not report numbers for these findings and will add them in the revised version.

---

> > ### Comment · Reviewer_D8Dy · 2021-08-24
> > **Response to reply**
> >
> > Thanks for the clarification that the Neurocoder is used for all layers of MLP. I believe I was misled by Figure 1 stating that "Here only the final layer of the Main Network is adaptively loaded...".
> >
> > Regarding $f_\varphi$, could you explicitly state its form for $\mathbf{M}_S$? The singular values are not discrete, so I don't think it would be one-hot. I share reviewer 4P5s's confusion about how the attention over singular values would work.

---

> > > ### Author Response · Authors · 2021-08-26
> > > **Implementation detail**
> > >
> > > Thank you for asking. Our example on one-hot vectors is to illustrate that we can use a key vector to represent the singular value (like a one-hot vector can be used to represent an integer). We did not mean the singular values had to be one-hot. The main point is we learn the representation mapping and hope that we can achieve useful representations for the singular vectors/values.
> > >
> > > Now we describe how we implement $f_{\varphi}$. Assume the size of memories $\mathbf{M_U}$, $\mathbf{M_V}$ and $\mathbf{M_S}$  are $P\times d_1$, $P\times d_2$, and $P\times 1$, respectively; and the key size is K. Then in this implementation, $f_{\varphi}$ is a neural network with an input size of $d_1+ d_2 + 1$ and output size of 3K (K<<d). $f_{\varphi}$ takes the concatenation (detached) [ $\mathbf{M_U}(i)$,  $\mathbf{M_V}(i)$, $\mathbf{M_S}(i)$] as input where i is the row index, and outputs 3 key vectors. We then perform attention to each of the 3 key vectors to get the attention weight to each slot in the memories (Eq 8).
> > > The weights are used to retrieve the components as in Eq. 3-5.
> > > The benefit of our attention formulation is:
> > >
> > > - Consistency. We do not need to design a special mechanism for singular value case. We apply the same mechanism to  $\mathbf{M_U}(i)$,  $\mathbf{M_V}(i)$, $\mathbf{M_S}(i)$.
> > > - Generic. Our formulation is flexible and can approximate straightforward mechanisms  generating singular values (see our reply to Reviewer 4P5s)
> > >
> > > We hope that this clarifies your remaining confusion.

---

### Author Response · Authors · 2021-08-09
**General reply**

Dear reviewers,

We appreciate your valuable feedback. Many of your comments are relevant and definitely will help us improve our paper. However, there remain misunderstandings that may hinder a proper evaluation. We will address these by replying to each of your reviews. We hope that our responses will address your concerns.

---

### Decision · Program_Chairs · 2021-09-28

**Decision:**

Reject

**Comment:**

The reviewers appreciate that this paper tackles an important problem of conditional computation, and proposed some interesting ideas.

All the reviews are slightly on the positive side, but some have low confidence.  Reading through the reviews, the AC recognizes that all reviewers were affected by the clarity issues of the paper, and there were quite a few misunderstandings about the model details, leading to perhaps inaccurate judgement of the paper.  This is in part caused by the fact that the model proposed in the paper is quite complicated with a lot of moving parts, which makes it challenging to present the model clearly.  On the other hand, only very limited ablations were done to understand the impact of each design decision.

The clarity issue and the absence of sufficient ablations are the two major factors holding back the AC’s decision to accept this paper.  After calibrating with SAC across a range of papers (we have both went through this paper ourselves) we decided to reject this paper but encourage the authors to improve it and send it to the next venue.

**Consistency Experiment:**

NeurIPS has a long history of experimentation. In 2014, NeurIPS ran an experiment in which 10% of submissions were reviewed by two independent committees to quantify the randomness in the review process. This year, we repeated a variant of this experiment to see how the quality of the review process has changed over time.  This paper was part of the experiment and was therefore assigned to two committees (consisting of reviewers, an Area Chair, and a Senior Area Chair) that reached independent decisions.  If both committees made the same recommendation, this recommendation was followed. If a single committee recommended acceptance, the paper was accepted (with the exception of a few cases in which the other committee identified what we considered a fatal flaw, e.g., an error in a key result).

Both committees reached the same decision: **Reject**

The other committee assigned to the paper recommended **Reject**.  You can find the other set of reviews, along with any follow up discussion with the authors here:
https://openreview.net/forum?id=jxASviAMAWD